# Body fat percentage is independently associated with lower pulmonary function in Korean never-smokers: A cross-sectional analysis of 33,748 adults

Eun Kyung Choe[1], Seung Ho Choi[2], Hae Yeon Kang [2]*

**1** Department of Surgery, Healthcare System Gangnam Center, Seoul National University Hospital, Seoul, Korea, **2** Department of Internal Medicine, Healthcare System Gangnam Center, Seoul National University Hospital, Seoul, Korea

* 65476@snuh.org

## Abstract

### Background

The relationship between obesity and pulmonary function is difficult to interpret using body mass index (BMI) alone because BMI cannot distinguish adiposity from lean mass. This study evaluated whether body fat percentage (BF%) is more strongly associated with pulmonary function than BMI in Korean never-smokers.

### Methods

We retrospectively analyzed 33,748 never-smoking adults who underwent spirometry and bioelectrical impedance analysis during health screenings between January 2007 and December 2014. Pulmonary function was assessed using forced vital capacity (FVC) and forced expiratory volume in one second ($FEV_1$) from pre-bronchodilator spirometry. Participants were stratified by BF% quartiles. Multivariable linear regression was used to evaluate the association between BF% and lung function after adjustment for age, BMI, waist circumference, and metabolic factors. In sensitivity analyses, lung function impairment was defined using lower limit of normal (LLN) criteria, and multivariable logistic regression was applied to evaluate restrictive and obstructive patterns.

### Results

Higher BF% was consistently associated with lower FVC and $FEV_1$ in both sexes across all BMI categories (all $P < 0.001$). After full adjustment, individuals in the lowest BF% quartile demonstrated significantly higher lung function than those in the highest quartile (β for FVC: +0.65 L in men and +0.36 L in women; β for $FEV_1$: +0.46 L in men and +0.28 L in women; all $P < 0.05$). These associations were most pronounced in

**Data availability statement:** Data Availability: Individual-level data cannot be shared publicly due to ethical restrictions imposed by the Institutional Review Board of Seoul National University Hospital, as the dataset contains potentially identifiable sensitive health information. These restrictions were imposed to protect participant privacy and confidentiality. De-identified individual-level data may be made available upon reasonable request to qualified researchers who meet the criteria for access to confidential data. Data access requests should be directed to the Institutional Review Board of Seoul National University Hospital (contact: https://hrpp.snuh.org/irb/introirb/_/singlecont/view.do) or to the corresponding author, who will facilitate the request process in accordance with institutional and ethical regulations. All aggregated data supporting the findings of this study are included within the manuscript and its Supporting information files.

**Funding:** The author(s) received no specific funding for this work.

**Competing interests:** The authors have declared that no competing interests exist.

younger men and in individuals with BMI ≥ 25 kg/m². In LLN-based analyses, elevated BF% was independently associated with a markedly increased risk of restrictive impairment in both men (OR 3.31, 95% CI 2.46–4.47) and women (OR 3.19, 95% CI 2.69–3.79), whereas no independent association was observed with obstructive patterns.

## Conclusion

BF% is independently and inversely associated with pulmonary function and is selectively linked to restrictive ventilatory impairment, offering more refined respiratory risk stratification than BMI in Korean never-smokers.

## Introduction

Obesity, defined by pathological adipose tissue accumulation, constitutes a major global health concern with established associations with metabolic, cardiovascular, and respiratory diseases [1–4]. Although Body Mass Index (BMI) remains the most widely used clinical metric for obesity, it has fundamental limitations in quantifying actual adiposity because it cannot distinguish between fat mass, lean mass, and skeletal components [5]. This limitation is particularly relevant in Asian populations, which tend to have higher body fat percentages and greater visceral adiposity at comparable BMI values than Western populations, contributing to a disproportionate burden of obesity-related diseases even at relatively modest BMI levels [6–9].

The relationship between BMI and pulmonary function has been inconsistent across studies and populations. While many investigations have reported inverse associations between increasing BMI and spirometric indices such as forced vital capacity (FVC) and forced expiratory volume in one second ($FEV_1$), findings for other lung volumes and across demographic subgroups have been variable, with some studies reporting neutral or even paradoxically favorable associations, particularly in women [10,11]. In addition, evidence suggests that the BMI–lung function relationship may be non-linear, with optimal respiratory function observed at intermediate BMI levels and deterioration at both underweight and obese extremes [12,13]. These heterogeneous findings underscore the limitations of BMI and the need for more physiologically relevant measures of adiposity in respiratory risk assessment.

Body fat percentage (BF%), assessed using techniques such as dual-energy X-ray absorptiometry (DXA) or bioelectrical impedance analysis (BIA), provides a more direct measure of total adiposity than BMI [14]. Despite these theoretical advantages, large population-based studies evaluating the association between BF% and pulmonary function remain limited, particularly in Asian populations where body composition differs substantially from that of Western cohorts. Furthermore, potential heterogeneity in BF%–lung function associations across demographic subgroups, including age, sex, and BMI categories, has not been systematically examined.

Accordingly, this study aimed to evaluate the independent association between BF% and pulmonary function in a large cohort of Korean never-smoking adults. The

specific objectives were to (1) compare the association of BF% with pulmonary function relative to BMI; (2) identify demographic subgroups in which BF%–pulmonary function associations are most pronounced; and (3) assess the potential clinical utility of body composition assessment for respiratory risk stratification. Although DXA and computed tomography (CT) are reference standards for body fat assessment, BIA was selected for its feasibility in large-scale health screening settings, and multi-frequency BIA has demonstrated strong correlation with DXA in Korean validation studies [15–17]. We hypothesized that BF% would show stronger and more consistent associations with pulmonary function than BMI, particularly in specific demographic subgroups, and that these associations would remain robust after adjustment for conventional risk factors.

## Methods

### Study design and population

We performed a retrospective cross-sectional analysis using data from the Seoul National University Hospital Healthcare System Gangnam Center, a major preventive health screening facility in Seoul, South Korea. The study included adults who underwent comprehensive health check-ups between January 2007 and December 2014. Of the 81,057 individuals initially screened, 33,748 never-smokers (8,327 men and 25,421 women) were eligible for analysis (S1 Fig). Participants were excluded if they had a history of malignancy, physician-diagnosed chronic pulmonary diseases (asthma, chronic obstructive pulmonary disease, or tuberculosis), missing or technically unacceptable spirometry or body composition data, or a history of current or former smoking. All included participants completed standardized questionnaires on medical history and lifestyle, physical examinations, laboratory testing, spirometry performed according to American Thoracic Society/European Respiratory Society guidelines [18], and body composition assessment using BIA.

### Data collection and measurements

All participants underwent standardized comprehensive health assessments conducted by experienced healthcare professionals using validated protocols. Detailed medical histories were obtained through structured questionnaires addressing medical conditions, medication use, lifestyle behaviors, and family history. Physical examinations and laboratory testing were performed concurrently with spirometry and body composition assessment to ensure data consistency.

Lifestyle factors were systematically assessed: regular exercise was defined as moderate- to high-intensity physical activity for ≥ 2 hours per week, and heavy alcohol consumption was defined as alcohol intake ≥ 140 g per week, based on established clinical criteria. Anthropometric measurements were obtained using standardized techniques. Height, weight, and BF% were measured using a validated multi-frequency BIA device (InBody 720; Biospace, Seoul, Korea). Waist circumference was measured at the midpoint between the lower costal margin and the iliac crest. BMI was calculated as weight (kg) divided by height squared (m²). Participants were categorized into BMI groups according to World Health Organization criteria adapted for Asian populations: underweight (< 18.5 kg/m²), normal weight (18.5–22.9 kg/m²), overweight (23.0–24.9 kg/m²), and obese (≥ 25.0 kg/m²) [19].

Laboratory assessments included fasting venous blood samples collected before 10:00 AM after an overnight fast of at least 12 hours. Measurements included fasting glucose, hemoglobin A1c (HbA1c), and a lipid profile (total cholesterol, triglycerides, low-density lipoprotein cholesterol, and high-density lipoprotein cholesterol). Diabetes mellitus was defined as fasting glucose ≥ 126 mg/dL, HbA1c ≥ 6.5%, or current use of antidiabetic medication. Hypertension was defined as systolic blood pressure ≥ 140 mmHg, diastolic blood pressure ≥ 90 mmHg, or current use of antihypertensive medication.

### Pulmonary function testing

Pre-bronchodilator spirometry was performed by experienced technicians in accordance with American Thoracic Society/European Respiratory Society guidelines [18]. A dry rolling seal spirometer (Model 2130, Viasys Respiratory Care, San

Diego, CA, USA) was used to measure FVC, $FEV_1$, and the $FEV_1$/FVC ratio. Results were expressed as absolute values (liters) and percent predicted values using reference equations derived from the Korean population [3,20]. Lower limits of normal (LLN) were applied to define spirometric impairment. Individualized LLN values for the $FEV_1$/FVC ratio were calculated for each participant using reference equations. Because Korean population–specific reference equations for individualized LLN estimation of $FEV_1$% and FVC% are not available, LLN thresholds for $FEV_1$% (81%) and FVC% (82%) were adopted from previously published large-scale Korean population studies [21,22]. Obstructive lung function was defined as $FEV_1$/FVC<LLN, and a restrictive spirometric pattern was defined as FVC%<LLN with $FEV_1$/FVC≥LLN [23].

## Statistical analysis

Statistical analyses were conducted using R version 3.2.2 (R Foundation for Statistical Computing, Vienna, Austria). Continuous variables are presented as means±standard deviations, and categorical variables as frequencies and percentages. All analyses were stratified by sex because of established physiological differences in body composition and pulmonary function between men and women.

Multivariable linear regression models were used to assess associations between BF% and pulmonary function parameters. BF% was analyzed both as a categorical variable (quartiles) and as a continuous variable. The fully adjusted model included age, BMI, waist circumference, lifestyle factors (exercise and alcohol consumption), and metabolic variables (diabetes mellitus, hypertension, triglycerides, and high-density lipoprotein cholesterol). Multivariable logistic regression was performed to evaluate the association between BF% quartiles and LLN-defined restrictive and obstructive spirometric patterns, with sequential adjustment for covariates. Subgroup analyses were conducted according to age (< 50 vs. ≥ 50 years) and BMI category. Statistical significance was defined as a two-sided P value<0.05.

## Ethics statement

The study protocol was approved by the Institutional Review Board of Seoul National University Hospital (IRB No. H-1606-095-771), with a waiver of informed consent granted because of the retrospective use of de-identified data. The study was conducted in accordance with the Declaration of Helsinki. Data were accessed for research purposes on 20 May 2024, and the authors had no access to identifiable participant information at any stage of the study.

## Results

### Participant characteristics

The final analytical cohort comprised 33,748 Korean never-smoking adults, including 8,327 men (24.7%) and 25,421 women (75.3%) (Table 1). Mean age was 48.2±11.3 years for men and 47.9±11.1 years for women. Mean BMI was 24.1±2.8 kg/m² in men and 22.4±3.0 kg/m² in women. BF% showed substantial variation across the cohort, with mean values of 22.3±4.1% in men and 29.2±4.9% in women, consistent with established sex differences in body composition. BF% quartile cut-offs were 19.3%, 22.3%, and 25.5% for men, and 25.7%, 29.2%, and 32.5% for women. Participants in higher BF% quartiles were older, shorter, and had higher BMI and waist circumference (all *P*<0.001), and a higher prevalence of metabolic conditions, including diabetes mellitus, hypertension, and dyslipidemia.

### Association between BF% and pulmonary function

Pulmonary function parameters by BF% quartile are presented in Table 2. Pulmonary function was consistently lower across increasing BF% quartiles in both men and women. In men, absolute FVC and $FEV_1$ values decreased from the lowest to the highest quartile (mean FVC: 4.5±0.7 L to 4.2±0.7 L; mean $FEV_1$: 3.8±0.6 L to 3.4±0.6 L). In women, absolute FVC decreased from 3.2±0.4 L to 2.9±0.5 L and $FEV_1$ from 2.8±0.4 L to 2.4±0.4 L across the same quartiles. Fig 1 illustrates FVC and $FEV_1$ across BMI categories and BF% quartiles, demonstrating lower pulmonary function with

**Table 1. Baseline characteristics of participants by sex and body fat percentage quartile.**

| | Men (n = 8,327) | | | | | Women (n = 25,421) | | | | |
|---|---|---|---|---|---|---|---|---|---|---|
| | Q1 (n = 2,082) | Q2 (n = 2,082) | Q3 (n = 2,082) | Q4 (n = 2,081) | *P* value | Q1 (n = 6,356) | Q2 (n = 6,355) | Q3 (n = 6,355) | Q4 (n = 6,355) | *P* value |
| Age, years | 41.9 ± 12.9 | 45.4 ± 12.0 | 46.4 ± 12.1 | 46.0 ± 12.8 | < 0.001 | 40.9 ± 10.1 | 43.6 ± 10.8 | 46.4 ± 11.2 | 50.6 ± 11.9 | < 0.001 |
| Regular exercise | 786 (37.8) | 689 (33.1) | 643 (30.9) | 555 (26.7) | < 0.001 | 1,978 (31.1) | 2,068 (32.5) | 1,987 (31.3) | 1,956 (30.8) | 0.154 |
| Heavy alcohol use | 342 (16.4) | 421 (20.2) | 481 (23.1) | 514 (24.7) | < 0.001 | 171 (2.7) | 162 (2.5) | 115 (1.8) | 137 (2.2) | 0.004 |
| Anthropometric measurements | | | | | | | | | | |
| Height, cm | 172.6 ± 5.9 | 171.1 ± 5.9 | 170.5 ± 5.9 | 169.4 ± 6.2 | < 0.001 | 160.8 ± 5.2 | 159.4 ± 5.1 | 158.2 ± 5.2 | 156.3 ± 5.4 | < 0.001 |
| Weight, kg | 64.7 ± 7.9 | 69.0 ± 7.6 | 72.2 ± 7.8 | 77.7 ± 10.6 | < 0.001 | 49.9 ± 4.8 | 53.1 ± 5.1 | 56.0 ± 5.5 | 61.6 ± 7.8 | < 0.001 |
| BMI, kg/m² | 21.7 ± 2.0 | 23.5 ± 1.8 | 24.8 ± 1.8 | 27.0 ± 2.7 | < 0.001 | 19.3 ± 1.5 | 20.9 ± 1.6 | 22.3 ± 1.8 | 25.2 ± 2.7 | < 0.001 |
| BMI < 23 | 1,527 (57.3) | 786 (29.5) | 286 (10.7) | 65 (2.4) | < 0.001 | 6,275 (36.3) | 5,715 (33.1) | 4,076 (23.6) | 1,221 (7.1) | < 0.001 |
| 23 ≤ BMI < 25 | 442 (17.4) | 868 (34.1) | 856 (33.7) | 376 (14.8) | < 0.001 | 76 (1.7) | 579 (13.0) | 1,815 (40.8) | 1,979 (44.5) | < 0.001 |
| BMI ≥ 25 | 113 (3.6) | 428 (13.7) | 940 (30.1) | 1,640 (52.5) | < 0.001 | 5 (0.1) | 61 (1.7) | 464 (12.6) | 3,155 (85.6) | < 0.001 |
| Waist circumference, cm | 79.1 ± 5.8 | 84.5 ± 5.0 | 88.1 ± 4.8 | 93.5 ± 6.8 | < 0.001 | 72.8 ± 5.1 | 77.3 ± 5.4 | 81.2 ± 5.7 | 87.9 ± 7.4 | < 0.001 |
| Body fat percentage, % | 16.3 ± 2.3 | 20.7 ± 0.9 | 23.8 ± 0.9 | 28.6 ± 2.8 | < 0.001 | 21.9 ± 2.3 | 26.7 ± 1.1 | 30.3 ± 1.0 | 35.3 ± 2.7 | < 0.001 |
| Hypertension | 160 (7.7) | 296 (14.2) | 379 (18.2) | 506 (24.3) | < 0.001 | 239 (3.8) | 455 (7.2) | 673 (10.6) | 1,315 (20.7) | < 0.001 |
| Diabetes mellitus | 72 (3.5) | 90 (4.3) | 101 (4.9) | 115 (5.5) | 0.012 | 96 (1.5) | 146 (2.3) | 185 (2.9) | 331 (5.2) | < 0.001 |
| Laboratory parameters | | | | | | | | | | |
| Total cholesterol, mg/dL | 182.1 ± 30.9 | 191.6 ± 32.8 | 194.6 ± 32.5 | 198.3 ± 33.7 | < 0.001 | 181.2 ± 31.1 | 188.0 ± 32.3 | 194.1 ± 34.2 | 201.8 ± 35.6 | < 0.001 |
| Triglycerides, mg/dL | 84.8 ± 44.9 | 106.9 ± 72.6 | 121.8 ± 69.8 | 137.8 ± 79.8 | < 0.001 | 67.6 ± 31.2 | 78.3 ± 44.4 | 88.8 ± 48.4 | 105.1 ± 57.9 | < 0.001 |
| LDL cholesterol, mg/dL | 110.6 ± 27.9 | 119.3 ± 30.2 | 121.8 ± 30.0 | 123.7 ± 31.1 | < 0.001 | 103.6 ± 27.3 | 111.7 ± 29.2 | 118.2 ± 31.1 | 125.3 ± 32.9 | < 0.001 |
| HDL cholesterol, mg/dL | 54.5 ± 11.6 | 50.9 ± 11.0 | 48.5 ± 10.0 | 47.1 ± 9.4 | < 0.001 | 64.1 ± 13.1 | 60.6 ± 13.1 | 58.2 ± 12.8 | 55.5 ± 12.2 | < 0.001 |
| Fasting glucose, mg/dL | 94.2 ± 15.0 | 96.5 ± 16.2 | 98.8 ± 17.9 | 101.3 ± 19.1 | < 0.001 | 87.7 ± 11.0 | 90.1 ± 13.0 | 92.3 ± 13.5 | 96.6 ± 17.7 | < 0.001 |
| Hemoglobin A1c, % | 5.6 ± 0.5 | 5.6 ± 0.6 | 5.7 ± 0.6 | 5.8 ± 0.7 | < 0.001 | 5.6 ± 0.4 | 5.6 ± 0.4 | 5.7 ± 0.5 | 5.8 ± 0.6 | < 0.001 |

Data are presented as mean ± standard deviation for continuous variables and n (%) for categorical variables. Body fat percentage (BF%) quartile cut-offs: Men: Q1 ≤ 19.3%, Q2 19.4–22.3%, Q3 22.4–25.5%, Q4 > 25.5%; Women: Q1 ≤ 25.7%, Q2 25.8–29.2%, Q3 29.3–32.5%, Q4 > 32.5%. P values compare BF% quartiles within each sex. Regular exercise was defined as moderate- to high-intensity physical activity ≥ 2 h/week. Heavy alcohol use was defined as alcohol intake ≥ 140 g/week.

Abbreviations: BF%, body fat percentage; BMI, body mass index; LDL, low-density lipoprotein; HDL, high-density lipoprotein.

increasing BF% across all BMI categories, including individuals with normal BMI. Across BMI groups, lower BF% was associated with relatively higher pulmonary function, including among participants classified as obese.

In multivariable-adjusted models, higher BF% was independently and inversely associated with both FVC and FEV$_1$ (Table 3). In the fully adjusted model (Model 3), men in the lowest BF% quartile had 0.65 L higher FVC (95% CI: 0.61–0.69) and 0.46 L higher FEV$_1$ (95% CI: 0.43–0.49) than those in the highest quartile (both *P* < 0.001). Corresponding differences in women were smaller but remained significant (FVC: + 0.36 L, 95% CI: 0.34–0.38; FEV$_1$: + 0.28 L, 95% CI: 0.26–0.30; both *P* < 0.001). These associations remained stable after adjustment for BMI, waist circumference, lifestyle factors, and metabolic conditions.

## Subgroup analyses by age and BMI

Subgroup analyses examining BF% as a continuous variable are shown in Table 4. BF% was inversely associated with pulmonary function across all age and BMI categories in both sexes. The strongest associations were observed in men younger than 50 years and in those with BMI ≥ 25 kg/m², in whom each 1% increase in BF% was associated with a 0.080 L decrease in FVC and a 0.057 L decrease in FEV$_1$ (*P* < 0.001). Women also demonstrated significant inverse associations,

**Table 2. Pulmonary function parameters by body fat percentage quartile, stratified by sex.**

| Variable | Total | Body fat percentage quartiles | | | | P for trend |
| --- | --- | --- | --- | --- | --- | --- |
| | | Q1 (lowest) | Q2 | Q3 | Q4 (highest) | |
| Men (n = 8,327) | | | | | | |
| Absolute values | | | | | | |
| FVC, L | 4.4 ± 0.7 | 4.5 ± 0.7 | 4.4 ± 0.7 | 4.3 ± 0.7 | 4.2 ± 0.7 | < 0.001 |
| FEV$_1$, L | 3.6 ± 0.6 | 3.8 ± 0.6 | 3.6 ± 0.6 | 3.5 ± 0.6 | 3.4 ± 0.6 | < 0.001 |
| FEV$_1$/FVC ratio, % | 82.6 ± 6.7 | 84.2 ± 7.3 | 82.1 ± 6.9 | 81.9 ± 6.5 | 82.4 ± 6.0 | < 0.001 |
| Percent predicted values | | | | | | |
| FVC, % predicted | 95.2 ± 11.1 | 95.4 ± 11.4 | 96.6 ± 11.0 | 95.3 ± 10.9 | 93.6 ± 10.8 | < 0.001 |
| FEV$_1$, % predicted | 103.6 ± 12.8 | 104.7 ± 12.5 | 104.6 ± 12.9 | 103.3 ± 12.7 | 101.7 ± 12.9 | < 0.001 |
| Women (n = 25,421) | | | | | | |
| Absolute values | | | | | | |
| FVC, L | 3.1 ± 0.5 | 3.2 ± 0.4 | 3.1 ± 0.4 | 3.1 ± 0.4 | 2.9 ± 0.5 | < 0.001 |
| FEV$_1$, L | 2.6 ± 0.4 | 2.8 ± 0.4 | 2.7 ± 0.4 | 2.6 ± 0.4 | 2.4 ± 0.4 | < 0.001 |
| FEV$_1$/FVC ratio, % | 84.6 ± 6.7 | 86.5 ± 6.9 | 84.9 ± 6.8 | 83.9 ± 6.5 | 82.9 ± 5.9 | < 0.001 |
| Percent predicted values | | | | | | |
| FVC, % predicted | 95.4 ± 12.3 | 92.9 ± 12.0 | 94.9 ± 12.4 | 96.2 ± 11.9 | 97.4 ± 12.1 | < 0.001 |
| FEV$_1$, % predicted | 105.0 ± 13.8 | 103.2 ± 12.9 | 104.3 ± 13.8 | 105.4 ± 13.5 | 107.1 ± 14.6 | < 0.001 |

Data are presented as mean ± standard deviation. Body fat percentage (BF%) quartile cut-offs: Men: Q1 ≤ 19.3%, Q2 19.4–22.3%, Q3 22.4–25.5%, Q4 > 25.5%; Women: Q1 ≤ 25.7%, Q2 25.8–29.2%, Q3 29.3–32.5%, Q4 > 32.5%. P for trend was calculated across BF% quartiles within each sex.

Percent predicted values calculated using Korean refrence equations. Abbreviations: FEV$_1$, forced expiratory volume in 1 second; FVC, forced vital capacity.

although the magnitude of effect was smaller. These findings indicate that excess body fat is associated with lower lung function independently of BMI, with particularly pronounced effects in younger men with higher BMI.

### Independent association of BF% with restrictive impairment

Multivariable logistic regression analyses demonstrated that higher BF% was independently associated with restrictive spirometric impairment in both sexes (Table 5). Compared with the lowest BF% quartile (Q1), the highest quartile (Q4) was associated with a more than three-fold increased odds of restrictive impairment in men (OR 3.312, 95% CI 2.460–4.469; P for trend < 0.001) and women (OR 3.190, 95% CI 2.685–3.790; P for trend < 0.001). Although the crude prevalence of restrictive patterns in women was higher in the lowest BF% quartile (Q1: 16.8% vs. Q4: 7.7%; S1 Table), this association was reversed after adjustment for age and BMI. In contrast, BF% was not independently associated with obstructive spirometric patterns (FEV$_1$/FVC ratio < LLN) in either sex after full adjustment.

### Discussion

This large-scale study of 33,748 Korean never-smokers demonstrates that BF% is independently associated with pulmonary function, particularly with restrictive ventilatory impairment. Elevated BF% showed consistent inverse associations with both FVC and FEV$_1$ across demographic subgroups, even after adjustment for BMI, waist circumference, and metabolic factors, supporting BF% as a physiologically relevant indicator of adiposity-related respiratory risk beyond BMI. These associations were most pronounced in younger men and individuals with BMI ≥ 25 kg/m², while clinically meaningful impairment was also observed in normal-BMI individuals with elevated BF%, highlighting an underrecognized high-risk group. Consistent with these findings, using LLN-based definitions, BF% was selectively associated with a restrictive

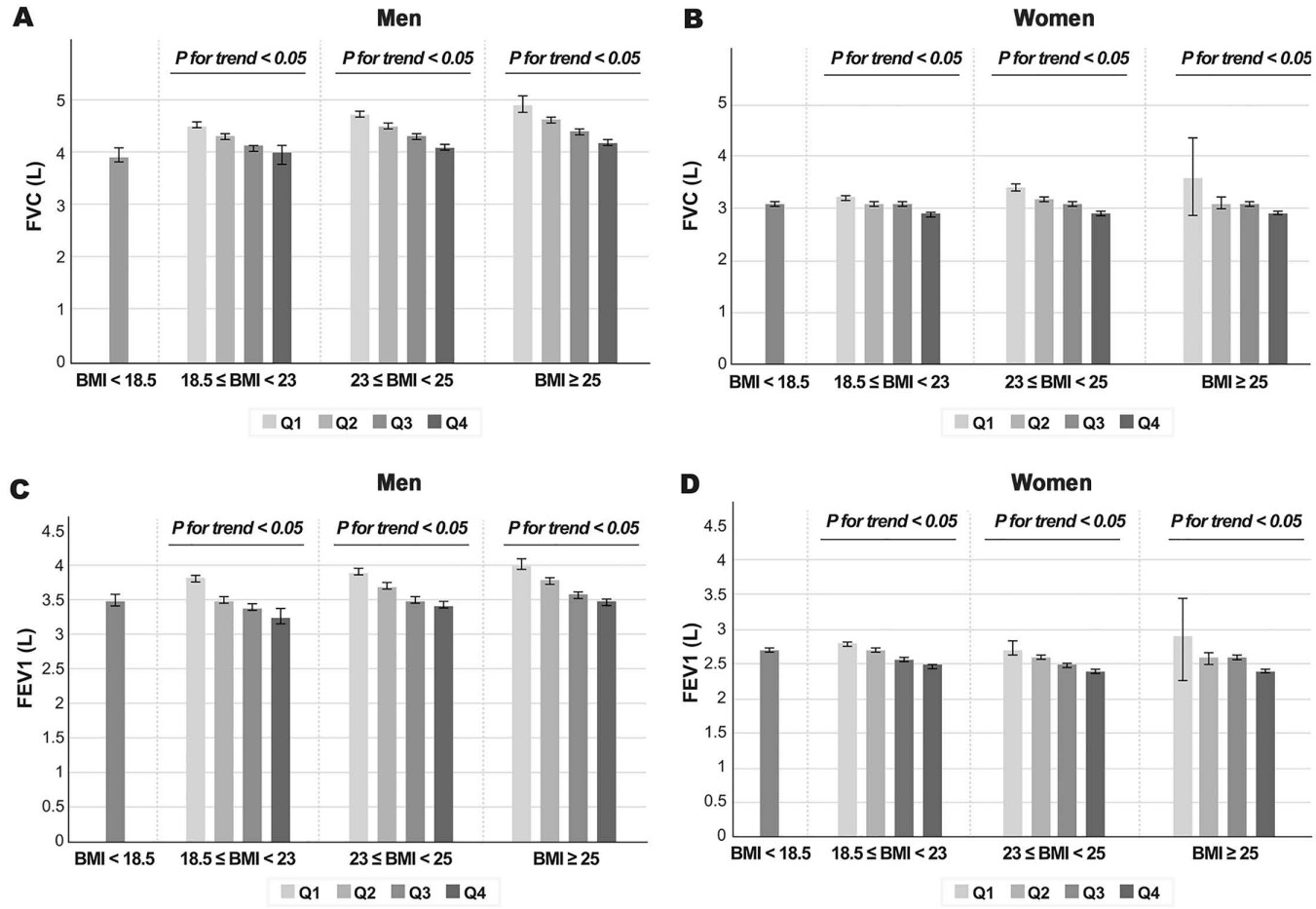

**Fig 1. Sex-stratified FVC and $FEV_1$ by BMI categories and BF% quartiles.** FVC and $FEV_1$ are shown according to BMI categories and BF% quartiles. Participants with BMI < 18.5 kg/m² were analyzed as a single group due to the small sample size. (A) FVC (L) in men; (B) FVC (L) in women; (C) $FEV_1$ (L) in men; (D) $FEV_1$ (L) in women. BF% quartile cutoffs were 19.3%, 22.3%, and 25.5% for men, and 25.7%, 29.2%, and 32.5% for women, defining Q1–Q4 from lowest to highest.

spirometric pattern, but not with obstructive impairment, in both men and women, reinforcing the interpretation that excess adiposity primarily affects lung volumes rather than airway caliber.

The clinical relevance of these findings is underscored by the magnitude of lung function differences observed. In men, a 6–8% difference in BF% between the lowest and highest quartiles was associated with reductions in $FEV_1$ of up to 0.46 L, corresponding to approximately 10–15 years of age-related decline in healthy, never-smoking adults (≈30–45 mL/year) [24,25]. The persistence of strong associations in LLN-based analyses, including a more than three-fold increased risk of restrictive impairment in the highest BF% quartile, further emphasizes the importance of excess adiposity as a determinant of reduced lung volumes independent of conventional obesity indices. From a public health perspective, these results support the inclusion of body composition assessment in obesity evaluation, particularly in Asian populations where BMI may underestimate adiposity-related risk [6,9].

Our findings extend prior studies that relied primarily on BMI and reported inconsistent associations with pulmonary function [26–29]. By directly quantifying total body adiposity, this study provides stronger evidence that BF% offers superior discrimination of pulmonary risk. The results are consistent with smaller Asian cohort studies [11,30], while adding

**Table 3. Multivariable linear regression of pulmonary function according to body fat percentage quartile, stratified by sex.**

| Variable | Q1 (lowest) | Q2 | Q3 | Q4 (highest) | P for trend |
|---|---|---|---|---|---|
| | β (95% CI) | β (95% CI) | β (95% CI) | Reference | |
| Men | | | | | |
| FVC, L | | | | | |
| Model 1 | +0.21 (0.17, 0.25) | +0.22 (0.18, 0.26) | +0.13 (0.09, 0.17) | 0 | < 0.001 |
| Model 2 | +0.66 (0.62, 0.70) | +0.50 (0.46, 0.54) | +0.30 (0.26, 0.34) | 0 | < 0.001 |
| Model 3 | +0.65 (0.61, 0.69) | +0.49 (0.45, 0.53) | +0.29 (0.25, 0.33) | 0 | < 0.001 |
| FEV$_1$, L | | | | | |
| Model 1 | +0.21 (0.18, 0.24) | +0.17 (0.14, 0.20) | +0.10 (0.07, 0.13) | 0 | 0.042 |
| Model 2 | +0.47 (0.44, 0.50) | +0.33 (0.30, 0.36) | +0.19 (0.16, 0.22) | 0 | 0.022 |
| Model 3 | +0.46 (0.43, 0.49) | +0.32 (0.29, 0.35) | +0.19 (0.16, 0.22) | 0 | 0.017 |
| Women | | | | | |
| FVC, L | | | | | |
| Model 1 | +0.11 (0.09, 0.13) | +0.10 (0.08, 0.12) | +0.08 (0.06, 0.10) | 0 | < 0.001 |
| Model 2 | +0.36 (0.34, 0.38) | +0.28 (0.26, 0.30) | +0.19 (0.17, 0.21) | 0 | < 0.001 |
| Model 3 | +0.36 (0.34, 0.38) | +0.27 (0.25, 0.29) | +0.19 (0.17, 0.21) | 0 | < 0.001 |
| FEV$_1$, L | | | | | |
| Model 1 | +0.12 (0.10, 0.14) | +0.08 (0.06, 0.10) | +0.06 (0.04, 0.08) | 0 | 0.009 |
| Model 2 | +0.28 (0.26, 0.30) | +0.20 (0.18, 0.22) | +0.13 (0.11, 0.15) | 0 | < 0.001 |
| Model 3 | +0.28 (0.26, 0.30) | +0.19 (0.17, 0.21) | +0.13 (0.11, 0.15) | 0 | < 0.001 |

Values are β coefficients (95% CI) from linear regression models, with Q4 (highest BF%) as the reference. Positive β coefficients indicate higher lung function compared with Q4. Body fat percentage (BF%) quartile cut-offs: Men: Q1 ≤ 19.3%, Q2 19.4–22.3%, Q3 22.4–25.5%, Q4 > 25.5%; Women: Q1 ≤ 25.7%, Q2 25.8–29.2%, Q3 29.3–32.5%, Q4 > 32.5%. Model 1: adjusted for age only. Model 2: adjusted for age, BMI, waist circumference. Model 3: adjusted for age, BMI, waist circumference, exercise, alcohol consumption, hypertension, diabetes mellitus, triglycerides, and high-density lipoprotein cholesterol.

Abbreviations: β, regression coefficient; CI, confidence interval; FEV$_1$, forced expiratory volume in 1 second; FVC, forced vital capacity; BMI, body mass index.

novel insights through comprehensive subgroup analyses and restriction to never-smokers. Younger men with elevated BMI and individuals with normal BMI but high BF% emerged as groups with particularly vulnerable respiratory profiles. Importantly, these findings were further strengthened by the use of an LLN-based approach, demonstrating that the observed associations are not driven by arbitrary percent-predicted cutoffs but instead reflect clinically meaningful reductions in lung volumes.

The mechanisms underlying the association between BF% and pulmonary function are likely multifactorial. Excess adiposity in the thoracic and abdominal regions restricts chest wall expansion and diaphragmatic excursion, leading to volume-dependent reductions in lung capacity [31,32]. The absence of an independent association between BF% and LLN-defined airflow obstruction further supports a predominantly mechanical effect rather than an airway-caliber–mediated process. In addition, visceral adipose tissue contributes to low-grade systemic inflammation through the secretion of cytokines and adipokines, including TNF-α, IL-6, leptin, and adiponectin, which may further impair pulmonary mechanics and respiratory muscle function [33,34].

Sex-specific patterns observed in this study warrant careful interpretation. In men, higher BF% was consistently associated with increased prevalence and risk of LLN-defined restrictive impairment, consistent with greater visceral fat accumulation and abdominal loading of the diaphragm. In women, the crude prevalence of restrictive patterns was higher in the lowest BF% quartile; however, this inverse association was reversed after multivariable adjustment, indicating substantial confounding by age, frailty, and reduced fat-free mass. After adjustment, high BF% emerged as an independent

**Table 4. Association between body fat percentage (continuous) and pulmonary function: subgroup analysis by age and BMI category.**

| Variable | 18.5 ≤ BMI < 23 kg/m² | | | 23 ≤ BMI < 25 kg/m² | | | BMI ≥ 25 kg/m² | | |
|---|---|---|---|---|---|---|---|---|---|
| | β | SE | P value | β | SE | P value | β | SE | P value |
| FVC, L | | | | | | | | | |
| All participants | −0.039 | 0.001 | < 0.001 | −0.045 | 0.002 | < 0.001 | −0.054 | 0.002 | < 0.001 |
| Men | −0.058 | 0.003 | < 0.001 | −0.063 | 0.004 | < 0.001 | −0.074 | 0.003 | < 0.001 |
| Men < 50 years | −0.060 | 0.004 | < 0.001 | −0.066 | 0.004 | < 0.001 | −0.080 | 0.004 | < 0.001 |
| Men ≥ 50 years | −0.052 | 0.006 | < 0.001 | −0.057 | 0.006 | < 0.001 | −0.060 | 0.005 | < 0.001 |
| Women | −0.036 | 0.001 | < 0.001 | −0.035 | 0.002 | < 0.001 | −0.034 | 0.002 | < 0.001 |
| Women < 50 years | −0.035 | 0.001 | < 0.001 | −0.035 | 0.003 | < 0.001 | −0.039 | 0.004 | < 0.001 |
| Women ≥ 50 years | −0.029 | 0.002 | < 0.001 | −0.030 | 0.003 | < 0.001 | −0.028 | 0.003 | < 0.001 |
| FEV₁, L | | | | | | | | | |
| All participants | −0.030 | 0.001 | < 0.001 | −0.033 | 0.002 | < 0.001 | −0.038 | 0.002 | < 0.001 |
| Men | −0.040 | 0.003 | < 0.001 | −0.044 | 0.003 | < 0.001 | −0.051 | 0.003 | < 0.001 |
| Men < 50 years | −0.038 | 0.004 | < 0.001 | −0.049 | 0.004 | < 0.001 | −0.057 | 0.003 | < 0.001 |
| Men ≥ 50 years | −0.041 | 0.005 | < 0.001 | −0.036 | 0.005 | < 0.001 | −0.037 | 0.004 | < 0.001 |
| Women | −0.027 | 0.001 | < 0.001 | −0.025 | 0.002 | < 0.001 | −0.025 | 0.002 | < 0.001 |
| Women < 50 years | −0.028 | 0.001 | < 0.001 | −0.026 | 0.003 | < 0.001 | −0.030 | 0.003 | < 0.001 |
| Women ≥ 50 years | −0.021 | 0.001 | < 0.001 | −0.022 | 0.002 | < 0.001 | −0.021 | 0.003 | < 0.001 |

Values are β coefficients (SE) representing the change in lung function (L) per 1% increase in BF%, estimated using multivariable linear regression. Negative β coefficients indicate lower lung function with increasing BF%. All models were adjusted for age, BMI, waist circumference, exercise, alcohol consumption, diabetes mellitus, hypertension, triglycerides, and high-density lipoprotein cholesterol.

Abbreviations: β, regression coefficient; BF%, body fat percentage; BMI, body mass index; FEV₁, forced expiratory volume in 1 second; FVC, forced vital capacity; SE, standard error.

determinant of restrictive impairment in women as well, with effect sizes comparable to those in men. These findings suggest that low BF% in women does not necessarily indicate preserved pulmonary health and underscore the importance of interpreting BF% in the context of age and body composition.

Although BMI and waist circumference are widely used in clinical practice, they remain imperfect proxies for true adiposity. BMI does not distinguish between lean and fat mass [35], and waist circumference largely reflects subcutaneous fat, whereas pulmonary function is more closely related to total and visceral adiposity [20,36]. The present findings demonstrate that BF% provides incremental and clinically meaningful information beyond conventional anthropometric indices, particularly for identifying individuals at risk of restrictive ventilatory impairment. Given the broad availability of BIA in health screening settings, routine BF% assessment may represent a pragmatic approach for early identification of individuals at elevated respiratory risk.

## Strengths and limitations

This study has several notable strengths. The large sample size provided substantial statistical power and enabled detailed stratified analyses by sex, age, and BMI category. Restriction to never-smokers minimized a major source of confounding in pulmonary function assessment, thereby strengthening internal validity. Body composition was assessed using validated bioelectrical impedance analysis, allowing direct evaluation of adiposity beyond BMI in a large-scale health screening setting. In addition, comprehensive adjustment for demographic, anthropometric, lifestyle, and metabolic factors enhanced the robustness and clinical relevance of the findings.

Several limitations should be acknowledged. The cross-sectional design precludes causal inference and limits assessment of temporal changes in adiposity and lung function. The study population consisted of self-referred health screening

**Table 5. Multivariable logistic regression analysis of impaired lung function by body fat percentage quartile, stratified by sex.**

| Variable | Men | | | | | | Women | | | | | |
|---|---|---|---|---|---|---|---|---|---|---|---|---|
| | Model 1 | | | Model 3 | | | Model 1 | | | Model 3 | | |
| | OR (95% CI) | P value | P for trend | OR (95% CI) | P value | P for trend | OR (95% CI) | P value | P for trend | OR (95% CI) | P value | P for trend |
| $FEV_1\% < LLN$ | | | <0.001 | | | <0.001 | | | 0.446 | | | <0.001 |
| Q1 | Reference (lowest) | | | Reference (lowest) | | | Reference (lowest) | | | Reference (lowest) | | |
| Q2 | 1.316 (0.878–1.989) | 0.186 | | 1.520 (0.992–2.349) | 0.056 | | 1.058 (0.863–1.298) | 0.587 | | 1.502 (1.210–1.865) | <0.001 | |
| Q3 | 1.431 (0.963–2.150) | 0.079 | | 1.812 (1.150–2.887) | 0.011 | | 0.961 (0.774–1.192) | 0.720 | | 1.856 (1.440–2.383) | <0.001 | |
| Q4 | 2.060 (1.422–3.032) | <0.001 | | 3.077 (1.857–5.151) | <0.001 | | 1.105 (0.886–1.377) | 0.375 | | 3.792 (2.775–5.173) | <0.001 | |
| FVC% < LLN | | | 0.008 | | | <0.001 | | | <0.001 | | | <0.001 |
| Q1 | Reference (lowest) | | | Reference (lowest) | | | Reference (lowest) | | | Reference (lowest) | | |
| Q2 | 0.840 (0.675–1.045) | 0.119 | | 1.201 (0.948–1.521) | 0.128 | | 0.886 (0.801–0.979) | 0.017 | | 1.375 (1.233–1.533) | <0.001 | |
| Q3 | 0.885 (0.712–1.099) | 0.269 | | 1.604 (1.238–2.080) | <0.001 | | 0.734 (0.658–0.818) | <0.001 | | 1.660 (1.458–1.890) | <0.001 | |
| Q4 | 1.294 (1.059–1.584) | 0.012 | | 3.372 (2.521–4.519) | <0.001 | | 0.735 (0.654–0.826) | <0.001 | | 3.365 (2.840–3.987) | <0.001 | |
| $FEV_1$/FVC ratio < LLN | | | 0.138 | | | 0.939 | | | 0.256 | | | 0.490 |
| Q1 | Reference (lowest) | | | Reference (lowest) | | | Reference (lowest) | | | Reference (lowest) | | |
| Q2 | 1.253 (0.902–1.744) | 0.180 | | 1.163 (0.822–1.649) | 0.394 | | 1.125 (0.929–1.362) | 0.227 | | 1.091 (0.894–1.333) | 0.392 | |
| Q3 | 1.331 (0.960–1.851) | 0.088 | | 1.177 (0.809–1.718) | 0.397 | | 1.095 (0.901–1.330) | 0.362 | | 1.055 (0.844–1.319) | 0.637 | |
| Q4 | 1.281 (0.922–1.782) | 0.140 | | 1.023 (0.650–1.612) | 0.922 | | 0.881 (0.714–1.087) | 0.238 | | 0.878 (0.653–1.177) | 0.385 | |
| Restrictive spirometric pattern | | | 0.012 | | | <0.001 | | | <0.001 | | | <0.001 |
| Q1 | Reference (lowest) | | | Reference (lowest) | | | Reference (lowest) | | | Reference (lowest) | | |
| Q2 | 0.823 (0.657–1.030) | 0.089 | | 1.174 (0.921–1.496) | 0.194 | | 0.867 (0.783–0.959) | 0.006 | | 1.340 (1.201–1.496) | <0.001 | |
| Q3 | 0.877 (0.702–1.096) | 0.249 | | 1.585 (1.216–2.069) | 0.001 | | 0.720 (0.645–0.804) | <0.001 | | 1.619 (1.420–1.847) | <0.001 | |
| Q4 | 1.279 (1.042–1.573) | 0.019 | | 3.312 (2.460–4.469) | <0.001 | | 0.705 (0.626–0.794) | <0.001 | | 3.190 (2.685–3.790) | <0.001 | |

Values are odds ratios (ORs) with 95% confidence intervals (CIs) from logistic regression models, with Q1 (lowest BF%) as the reference. BF% quartile cut-offs: Men: Q1 ≤ 19.3%, Q2 19.4–22.3%, Q3 22.4–25.5%, Q4 > 25.5%; Women: Q1 ≤ 25.7%, Q2 25.8–29.2%, Q3 29.3–32.5%, Q4 > 32.5%. P for trend was calculated across BF% quartiles. Model 1: adjusted for age only. Model 3: adjusted for age, BMI, waist circumference, exercise, alcohol consumption, hypertension, diabetes mellitus, triglycerides, and high-density lipoprotein cholesterol.

LLN for the $FEV_1$/FVC ratio was calculated for each participant using reference equations. LLN thresholds for FVC% and $FEV_1$% were 82% and 81%, respectively, based on a previous large-scale Korean study. Airflow limitation (obstructive impairment) was defined as $FEV_1$/FVC ratio < LLN. A restrictive spirometric pattern was defined as FVC% < LLN with $FEV_1$/FVC ratio ≥ LLN. Abbreviations: CI, confidence interval; BF%, body fat percentage; $FEV_1$, forced expiratory volume in 1 second; FVC, forced vital capacity; LLN, lower limit of normal; OR, odds ratio.

participants and may not fully represent the general Korean population. Direct lung volume measurements, such as total lung capacity and residual volume, were unavailable, limiting differentiation between intrinsic restrictive lung disease and obesity-related mechanical restriction. Although a hybrid LLN-based approach was applied, the absence of Korean-specific reference equations for individualized LLN estimation of $FEV_1\%$ and FVC% remains a limitation, highlighting the need for future validation studies. While BIA enabled standardized assessment in this large cohort, it is less precise than gold-standard imaging modalities, despite reasonable concordance with DXA reported in Korean populations. Finally, unmeasured factors such as diet and socioeconomic status may have contributed to residual confounding, and restriction to never-smokers limits generalizability to smoking populations.

## Conclusion

In this large cross-sectional study of Korean never-smokers, higher BF% was independently associated with lower pulmonary function and a substantially increased risk of restrictive ventilatory impairment, even after adjustment for BMI, waist circumference, and metabolic factors, with the strongest associations observed in younger men and individuals with elevated BMI. Importantly, impaired pulmonary function was also evident among normal-BMI individuals with high BF%, identifying a high-risk group not captured by BMI alone. LLN-based analyses further demonstrated that excess adiposity was selectively linked to reduced lung volumes rather than airflow obstruction.

Overall, BF% emerges as a clinically informative and physiologically relevant marker for respiratory risk stratification. Incorporating body composition assessment into routine evaluation may improve early identification of individuals vulnerable to adiposity-related pulmonary impairment, particularly in Asian populations, and underscores the need for longitudinal studies to clarify causality and potential reversibility.

## Supporting information

**S1 Fig. Flow diagram of study participant selection.**
(PDF)

**S1 Table. Impaired lung function according to body fat percentage quartiles, stratified by sex.**
(DOCX)

## Author contributions

**Conceptualization:** Eun Kyung Choe, Hae Yeon Kang.

**Data curation:** Eun Kyung Choe, Hae Yeon Kang.

**Formal analysis:** Eun Kyung Choe, Hae Yeon Kang.

**Investigation:** Eun Kyung Choe, Seung Ho Choi, Hae Yeon Kang.

**Methodology:** Eun Kyung Choe, Seung Ho Choi, Hae Yeon Kang.

**Resources:** Eun Kyung Choe, Hae Yeon Kang.

**Supervision:** Hae Yeon Kang.

**Writing – original draft:** Eun Kyung Choe, Hae Yeon Kang.

**Writing – review & editing:** Seung Ho Choi, Hae Yeon Kang.

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
