## [Decision Letter · Decision Letter 0]

5 Nov 2025

Dear Dr. Kang,

While the reviewers recognize the merit of your dataset and the potential contribution of your findings, addressing the above concerns is essential to improve the scientific rigor, clarity, and reproducibility of your work.

A detailed **point-by-point response letter** , indicating how each comment has been addressed.A **marked (tracked changes)** version of the revised manuscript.A **clean version** for publication consideration.

Thank you for considering PLOS One for your work.

Please submit your revised manuscript by Dec 20 2025 11:59PM . If you will need more time than this to complete your revisions, please reply to this message or contact the journal office at plosone@plos.org . A rebuttal letter that responds to each point raised by the academic editor and reviewer(s). You should upload this letter as a separate file labeled 'Response to Reviewers'.A marked-up copy of your manuscript that highlights changes made to the original version. You should upload this as a separate file labeled 'Revised Manuscript with Track Changes'.An unmarked version of your revised paper without tracked changes. You should upload this as a separate file labeled 'Manuscript'.

We look forward to receiving your revised manuscript.

Kind regards,

Khadija Ayed, MD, Ph.D

Academic Editor

PLOS ONE

**Journal Requirements:**

Reviewers' comments:

Reviewer's Responses to Questions

**Comments to the Author**

1. Is the manuscript technically sound, and do the data support the conclusions?

Reviewer #1: Yes

Reviewer #2: Yes

2. Has the statistical analysis been performed appropriately and rigorously?

Reviewer #1: Yes

Reviewer #2: Yes

3. Have the authors made all data underlying the findings in their manuscript fully available?

Reviewer #1: Yes

Reviewer #2: Yes

4. Is the manuscript presented in an intelligible fashion and written in standard English?

Reviewer #1: Yes

Reviewer #2: Yes

**Reviewer #1:** The authors report interesting data on the effects of BF, assessed by BIA, on lung function testing in a large Korean cohort based on retrospective data.

To improve the manuscript following issues can be addressed by the authors:

1. Definition of lung function impairment. The authors must be more precise in the definitions. They used % pred for FEV1 and FVC and a standard cut-off for FEV1/FVC ratio. The authors could perform a sensitivity analysis based on LLN. Furthermore, besides flow limitation, the authors can consider the presence of restrictive spirometry.

2. Discussion. The authors formulate standard principles about the role of BF but do not bring these principles in relation to their findings. For example, BF as endocrine organ: what about the highest lung function impairment in women with lowest BF paurtile? ( line 184).

3. A major limitation of this retrospective study is the absence of lung volume measurements. The authors must address this limitation.

**Reviewer #2:** 1. The study is cross-sectional, but some interpretations suggest causality (e.g., “predicts pulmonary function decline”). My suggestion is to revise throughout to avoid causal language. Replace “predicts” with “is associated with” unless longitudinal data are included.

2. Discuss potential measurement error or variability and why BIA was chosen over DXA

3. Physical activity, dietary factors, and socioeconomic status could confound both adiposity and pulmonary outcomes. Acknowledge these unmeasured confounders as a limitation

4. Terms like “pulmonary function decline” imply longitudinal worsening; since this is cross-sectional, it should read “lower pulmonary function”.

5. Data collection time duration not mentioned

6. Specify whether lung function was pre- or post-bronchodilator

7. Add reference for BIA validation in Korean populations.

8. Clarify definition of “impaired pulmonary function” (thresholds used).

9. Figures are clear but could include 95% confidence intervals to improve interpretability

**Do you want your identity to be public for this peer review?** For information about this choice, including consent withdrawal, please see our Privacy Policy

Reviewer #1: No

Reviewer #2: **Yes:**  Dhiraj Agarwal

---

## [Author Response · Author response to Decision Letter 1]

22 Dec 2025

We sincerely thank the reviewers for their constructive comments, which substantially improved the clarity and rigor of our manuscript. In response, we refined the definition of impaired pulmonary function using lower limit of normal (LLN)–based criteria by applying individualized LLN values for the FEV₁/FVC ratio and validated Korean surrogate LLN thresholds for FEV₁% and FVC%. Based on these criteria, restrictive and obstructive spirometric patterns were newly defined and incorporated into additional sensitivity analyses, with results presented in newly added tables. The revised analyses clarified that apparent sex-specific differences in the original submission were largely due to confounding, with consistent associations between body fat percentage and restrictive ventilatory impairment observed in both men and women after full adjustment. We also revised the manuscript to avoid causal language, expanded discussion of key limitations, clarified spirometry methods and data collection periods, and updated figures to include 95% confidence intervals. A detailed, point-by-point response to each reviewer comment is provided in a separate attached file.

---

## [Editor Report · Decision Letter 1]

14 Jan 2026

Body fat percentage is independently associated with lower pulmonary function in Korean never-smokers: A cross-sectional analysis of 33,748 adults

PONE-D-25-50052R1

Dear Dr. Hae Yeon Kang,

We’re pleased to inform you that your manuscript has been judged scientifically suitable for publication and will be formally accepted for publication once it meets all outstanding technical requirements.

Kind regards,

Khadija Ayed, M.D., Ph.D

Academic Editor

PLOS One

---

## [Editor Report · Acceptance letter]

PONE-D-25-50052R1

PLOS One

Dear Dr. Kang,

I'm pleased to inform you that your manuscript has been deemed suitable for publication in PLOS One. Congratulations! Your manuscript is now being handed over to our production team.

Kind regards,

on behalf of

Dr. Khadija Ayed

Academic Editor

PLOS One